# Exploring empathic engagement in immersive media: An EEG study on mu rhythm suppression in VR

**Jong-Hyun Lee[1], Sung Eun Lee[2], Young-Sung Kwon** [ID][3]*

1 Brain and Humanity Lab., Institute of Humanities, Seoul National University, Seoul, South Korea,
2 Department of German Language & Literature, Seoul National University, Seoul, South Korea,
3 Department of Media & Communication, Dong-A University, Busan, South Korea

* yoskwon@dau.ac.kr

## Abstract

This study investigates the influence of immersive media, particularly Virtual Reality (VR), on empathic responses, in comparison to traditional television (TV), using electroencephalography (EEG). We employed mu rhythm suppression as a measurable neural marker to gauge empathic engagement, as its increase generally signifies heightened empathic responses. Our findings exhibit a greater mu rhythm suppression in VR conditions compared to TV conditions, suggesting a potential enhancement in empathic responses with VR. Furthermore, our results revealed that the strength of empathic responses was not confined to specific actions depicted in the video clips, underscoring the possibility of broader implications. This research contributes to the ongoing discourse on the effects of different media environments on empathic engagement, particularly emphasizing the unique role of immersive technologies such as VR. It invites further investigation into how such technologies can shape and potentially enhance the empathic experience.

## Introduction

Empathy, one of key elements in deepening audience engagement, plays a significant role in enhancing the overall media consumption experience [1, 2]. It is defined as the ability to understand and share the feelings of others, leading audiences to emotionally or cognitively connect with characters by experiencing their joy, sorrow, and excitement, or by understanding their thoughts, motivations, and decisions [3, 4]. Such empathic connections with the characters can promote emotional investment and foster a more profound comprehension of the characters' viewpoints, deepening audiences' engagement with the narrative [1–7]. This engagement through empathy or empathic engagement might result in an enhanced media experience, blurring the lines between the viewer and the character, thus enriching the audience's experience and expanding their appreciation for the diverse perspectives and experiences presented in the media.

Empathy has been extensively studied in relation to immersion, as immersive experiences have been shown to have a significant impact on eliciting empathy [8–19]. Immersion deeply

**Data Availability Statement:** Data(including EEG) are fully available without restriction from the corresponding author upon request. For the long term stability and availability of data, data are also available from the University hospital IRB(+82-51-

200-6503/irb@dau.ac.kr/https://dms.donga.ac.kr/research2/index.do).

**Funding:** This work was supported by the National Research Foundation of Korea (NRF) grant funded by the Korea government awarded to Y-SK [MSIT; No. NRF-2020R1G1A1101384] and the Ascending SNU Future Leader Fellowship through Seoul National University awarded to J-HL. The funders had no role in study design, data collection and analysis, decision to publish, or preparation of the manuscript.

**Competing interests:** The authors have declared that no competing interests exist.

captivates the audience, drawing them into a lifelike and compelling experience. This intense captivation, by making the story world more tangible and relatable, fosters a stronger empathetic connection with the characters and their journeys [20]. For instance, in cinema, the use of surround sound, large screens, and 3D technology can envelop viewers in the movie's audio landscape and make visual elements more lifelike, which helps viewers to feel physically closer to the action. This allows viewers to feel more involved in the narrative, which in turn fosters empathic engagement [21]. Similarly, in literature, well-crafted descriptive language and engaging narrative techniques can transport readers into the story world, evoking strong empathic responses towards the characters and their experiences [22]. This suggests the degree of immersion may play a critical role in enhancing empathic responses in media experiences, making it an essential factor to consider when exploring the potential of different media formats to elicit empathy.

VR technology is becoming increasingly recognized as a device specialized for creating such immersive experiences. It generates a deep sense of presence and immersion by visually isolating users from their real-world environments [23], and enhancing the sensation of being present in a virtual space [24]. As an immersive device, VR introduces unique possibilities for media consumption [25], integrating these immersive experiences into various aspects of media, from video games to cinema to sports and even to rehabilitation [26].

This immersive quality of VR distinguishes it from traditional forms of media such as TV, potentially enabling a deeper, more nuanced empathic experience for audiences. In fact, building on its immersive qualities, VR has even been described as the 'ultimate empathy machine' [27]. The immersive nature of VR allows users to not only observe but also virtually embody the experiences and perspectives of others, facilitating a deeper understanding of their emotions and motivations. For instance, the VR film *Clouds Over Sidra* [28], which follows the life of a Syrian girl in a refugee camp, has been praised for its capacity to evoke empathy and understanding for the plight of refugees, without using special dramatic techniques [20].

However, empirical evidence supporting the effectiveness of VR in inducing empathy is still accumulating, and some researchers have reported mixed results [9, 12, 13, 15, 17, 19, 29–31]. For instance, the extent to which VR elicits empathetic experiences compared to traditional media may vary, depending on factors such as users' individual characteristics, their viewing perspective, and the specific task at hand [13, 17, 29]. Furthermore, neurophysiological evidence, such as EEG or fMRI data, remains scarce in this domain [32, 33], leaving a critical gap in our understanding of the underlying neural mechanisms involved in VR-induced empathic experiences. Consequently, further research is needed to better comprehend the true potential of VR as an empathy-enhancing medium and validate its claim as the 'ultimate empathy machine'.

From a neurobiological perspective, empathy is often closely tied to the activation of the mirror neuron system. This intriguing neural network, initially identified in primates [34] and subsequently demonstrated in humans [35], is thought to be central to the capacity of experiencing empathy [36, 37]. The foundational principle behind this association is that mirror neurons are activated not just when an individual performs an action but also when they observe the same action being carried out by another [37, 38]. For instance, when participants observed simple hand movements, such as grasping, the same brain areas were activated as when they performed these movements themselves [37]. This dual activation creates a form of neural mirroring or resonance, which essentially allows our brains to mimic or replicate the observed action in internal neural systems, providing a potential neurological basis for the ability to understand and share the emotions and experiences of others, a fundamental aspect of empathy [37, 39, 40]. This distinctive feature of mirror neurons, hence, establishes the fundamental connection between the neurons and the capacity for empathy.

One of the most common neural correlates associated with the activation of the mirror neuron system is the suppression of the mu rhythm, a unique EEG pattern typically observed over the sensorimotor cortex [41]. The mu rhythm is noted to be suppressed, or 'desynchronised', during both action execution and action observation, mirroring the behaviour of the mirror neurons themselves [41, 42]. Thus, the suppression of the mu rhythm is an indirect measure of mirror neuron activity and, by extension, a quantifiable indicator of empathic responses [43, 44]. Research has indicated the magnitude of mu suppression during action observation may be reflective of the observer's empathic engagement [44–46]. For example, Perry et al. [46] found a correlation between the degree of mu suppression and the extent of empathy an observer feels for the person performing the action. Woodruff et al. [44] demonstrated individuals who exhibited stronger mu suppression also reported higher levels of empathy. These studies suggest mu rhythm suppression serves not only as a neural marker of mirror neuron system activity but also as a potential measure of empathic engagement.

Building on this background, in the present study, we aim to acquire a deeper understanding of how the immersive nature of VR can influence empathic responses in contrast to traditional TV, utilising EEG methodologies. The central inquiry of this research asks whether the immersive characteristics of VR augment the empathic experience in comparison to TV. In an attempt to answer this, we measure the degree of mu rhythm suppression, an EEG pattern associated with empathy, because it provides a glimpse into the workings of the mirror neuron system. During the EEG experiment, participants will be presented with a series of short video clips that display human actions in both VR and TV settings, which allows for a direct comparison of empathic responses triggered by each medium. The EEG activity of participants will be recorded during viewing to quantify the degree of mu rhythm suppression, thereby providing empirical evidence to assess the magnitude of empathic responses. This study includes two types of actions of varying complexity: simple, such as object grasping, and complex, such as punching and kicking. Although object grasping is typically used in action observation studies [34, 47–49], its simplicity might limit one's ability to fully represent empathetic responses. Thus, in this study, by probing the impact of action complexity on these responses, we attempt to indirectly assess the representativeness of this simple task. If the mu rhythm suppression is only triggered by actions in the simple conditions, this could suggest these reactions are specific to certain actions, which may limit our understanding of empathetic responses in more intricate narrative scenarios. Otherwise, the findings might suggest a potential applicability of the study's results beyond the specific experimental items. The guiding hypothesis of this study is that the immersive VR environment will stimulate a stronger empathic response, indicated by a greater degree of mu rhythm suppression, compared to the TV setting. Furthermore, the complexity of actions depicted in the video clips might modulate mu rhythm suppression across VR and TV conditions, if the responses are limited to a certain type of actions.

## Methods

### Participants

We recruited a total of 30 participants (21 male and nine female university students, all of them were South Koreans) from the university's website (Recruited from December 2022 to January 2023), with a mean age of 24.10 years (SD = 2.86). They were all right-handed, with normal or corrected-to-normal vision, and had some experience with VR in the past Prior to the experiment, we obtained written informed consent from each participant. We conducted the experiment in accordance with the guidelines approved by university hospital Institutional Review Board (IRB Approval number: 2-1040709-AB-N-01-202202-HR-016-04). Data

(including EEG) are available from the corresponding author and University hospital IRB upon request.

## Materials

The experimental stimuli comprised 60 short video clips, which were used identically in both media Each was shot with the Insta EVO 360 and then edited using Adobe Premiere Pro. Actors who appeared in the stimulus were recruited from aspiring actors at the university. The stimuli were classified into two categories based on the complexity of the depicted actions: simple and complex. Simple actions encompassed the grasping of an object such as a bottle, cup, or ball, with two variations: either halting the movement postgrasp (no withdrawal) or grasping and subsequently moving the object out of the screen's frame (withdrawal). This grasping movement is a basic action that has been widely employed in several earlier studies [34, 47–49]. Complex actions included a range of activities performed by an actor, such as punching and kicking. These actions are distinct from simple actions in that they encompass movements of the entire body rather than being confined to a right arm and have two or more interconnected motions. They were adapted from the previous studies investigating how the mirror neuron system is modulated based on the level of complexity of actions [50–56]. In the videos, actors were shown both in full and partially, depending on the action being performed. For simpler actions, a closer, more focused shot was used. For complex actions involving whole-body movement, a wider shot was employed to capture the entire range of motion. Each video clip had a duration of 5 s, preceded by a 5-s black screen, thus totalling a 10-s duration per trial. Using a Latin-square design, we presented these materials in a random and counter-balanced manner. To account for potential order effects, we randomly assigned participants into two groups: one group first experienced the TV condition followed by the VR condition, whereas the other group underwent the conditions in reverse order. This design ensured a balanced representation of the TV and VR conditions across the participant pool.

## Procedure

Upon arrival at the laboratory, we provided the participants with instructions and a consent form. After agreeing to participate, they were seated in a noise-attenuated shield room. After agreeing to participate, they were seated in a noise-attenuated shield room. They were then instructed to attentively observe the brief video segments, focusing on the various movements displayed. While they were watching the clips, we collected their EEG responses. Each experimental session, whether it was the TV or VR condition, lasted approximately 10 min. After completing one condition, we gave the participants a break before proceeding to the next condition while the video device was switched. The total duration of the experiment, inclusive of preparation time and breaks, ranged from 30–40 min. Participants were compensated with 30,000 KRW (equivalent to approximately 24 USD) for their participation.

## Apparatus

For the TV condition, participants viewed the clips from a distance of 60 cm on a 24-in. monitor. In the VR condition, participants watched the video clips using an Oculus Rift S VR headset. The position of the headset was adjusted as needed to ensure participant comfort. We used virtual desktop applications (Virtual Space) to execute the VR condition and played the clips using a Python script.

## EEG recording

EEG data were recorded using a 16-channel actiCAP Xpress V-amp EEG recorder (Brain Products). The data were digitized at a sampling rate of 500 Hz. Given that the actiCAP Xpress system allows for high impedance, the stability of the recording quality was visually assessed after achieving a minimum level of impedance. The dry electrode system, by means of this recording procedure, can reliably yield EEG spectra and ERP components comparable to those obtained from traditional electrode types [57]. Electrodes were placed according to the international 10–20 system, with the reference electrode positioned at the right earlobe and the ground electrode at the left earlobe. The EEG data were filtered online using a low cut-off of 0.1 Hz.

To accommodate both the VR and EEG devices, participants were first fitted with the EEG device and the signal quality was stabilized. Subsequently, the VR device was placed on the participant. Since the VR headset was designed without any attachments above the upper part of the head (dorsal area), it is possible to ensure minimal overlap with the EEG device. Once the VR and EEG devices were in place, the signal quality was checked again to confirm stability.

**EEG preprocessing.**  All pre-processing of the raw EEG signals was conducted using the MNE-python package [58]. The raw EEG signals were first notch-filtered at 60 Hz and band-pass filtered within a frequency range of 1 to 40 Hz using a one-pass, zero-phase, non-causal FIR filter (Hamming window method, -6 dB cutoff frequencies: 0.50 Hz and 45.00 Hz). This initial step served to minimize noise and to focus on the most pertinent frequency range for the EEG signals. Following the filtering stage, ocular artifacts were corrected through an Independent Component Analysis (ICA) using the "picard" algorithm [59]. After the ICA ocular correction, the data were segmented into epochs spanning from 1 second before stimulus onset to 5 seconds after. The baseline was defined as the interval from -1 to 0 second relative to stimulus onset. The Python package 'autoreject' [60] was used for artifact rejection, which provides a global rejection threshold and interpolates bad sensors for each epoch. The total data loss as a result of this artifact rejection process amounted to 6.47% (5.22% for TV, 7.72% for VR). After the artifact rejection, the data from two participants were excluded from further analysis because more than 30% of their trials in one of the two media conditions were rejected due to artifacts, in order to ensure a high-quality dataset for subsequent analysis. As a result, the final analysis included data from 28 participants.

Time-frequency analysis was conducted using a Morlet wavelet transform, set to seven cycles, with the frequency of interest ranging from 4 Hz to 30 Hz at 1 Hz intervals. The analysis focused on electrodes C3 and C4, which are commonly associated with the sensori-motor cortex and the mu rhythm suppression [56, 61–63]. Power values were normalized to a 'percent' rescale mode using a one-second pre-stimulus baseline period of black screen presentation. Upon obtaining the power for each epoch, these values were then averaged within each condition separately, yielding an average power value for each condition at each time-frequency point.

**Statistical analysis.**  Statistical analyses were conducted using two primary methods: a repeated measures ANOVA and a non-parametric permutation F-test. The repeated measures ANOVA was performed using the rstatix library in R [64], which targeted the mu rhythm range (8 to 13 Hz) during a time window from stimulus onset to 5 seconds post-stimulus across two EEG channels, C3 and C4. The main effects of media (VR vs. TV) and complexity (simple vs. complex) were examined, as well as their interaction. The assumption of sphericity was checked, and if violated, the Greenhouse-Geisser correction was applied. A Bonferroni correction was used to address multiple comparisons, setting the significance level at 0.025 (0.05 divided by the number of channels, which is 2). In the event of any significant

interactions, post-hoc pairwise comparisons would have been conducted. For the non-parametric permutation F-test, a similar analysis was conducted using the MNE-Python package. As with the ANOVA, the analysis targeted the C3 and C4 channels, but examined a frequency range of 8 to 30 Hz at 1 Hz intervals within the same time window of 0 to 5 seconds post-stimulus onset. The permutation test was based on 1024 permutations, and clusters were defined as adjacent time-frequency points where the observed F-value exceeded the 95th percentile of the permutation distribution ($p < 0.05$). The cluster-level test statistic was the sum of the F-values within each cluster. To correct for multiple comparisons (two channels), a cluster-level threshold of $p < 0.025$ was used (Bonferroni correction), meaning that only clusters with a p-value less than 0.025 were considered statistically significant. Two separate tests were conducted: one comparing the two media conditions (VR vs. TV), and the other comparing the two complexity conditions (simple vs. complex).

## Results

### ANOVA

The repeated measures ANOVA revealed a significant main effect of media on both C3 and C4 channels, with no other significant main effects or interactions observed (C3: $F(1,27) = 25.331$, $p < 0.001$, C4: $F(1,27) = 22.515$, $p < 0.001$). In the VR conditions, a larger suppression of the mu rhythm was observed across both channels compared to the TV conditions (Figs 1 and 2).

### Non-parametric permutation F-test

The non-parametric permutation F-test results were consistent with the ANOVA findings, as a significant cluster ($p < 0.001$) emerged within the mu rhythm range (8 to 13 Hz) during the media analysis (VR vs TV). The VR condition exhibited a more pronounced suppression compared to TV (Fig 3). As for the complexity analysis (Simple vs Complex), no significant cluster was identified in the mu range, but a significant cluster ($p < 0.025$) was found in the beta frequency range (15 to 25 Hz) from 3 to 4 seconds. In this cluster, the complex condition displayed higher beta suppression than the simple one (Fig 4).

## Discussion

In this study, we sought to investigate the impact of immersive media, specifically VR, on empathic experiences in comparison to traditional TV, utilising EEG as a measure of mu rhythm suppression, a neural marker associated with empathy. The results revealed that, both in the ANOVA and the nonparametric permutation F-test, there was a greater suppression of the mu rhythm in the VR condition compared to the TV condition, over central electrodes, suggesting VR elicits a stronger empathic response. Moreover, there was no significant interaction observed between the media type and the complexity of actions depicted in the video clips in both analyses. Finally, the nonparametric permutation F-test indicated there existed a significant difference in beta power between the simple and complex conditions, albeit not in the mu rhythm range, with the complex condition showing higher beta suppression.

The observed difference in mu rhythm suppression between the VR and TV conditions in our study implies a distinct activation of the mirror neuron system, commonly associated with empathic responses. This aligns with previous studies whose authors reported a heightened emotional engagement when interacting with VR content, as gauged through self-report surveys [9, 13, 17, 29, 30] and behavioural indices [15, 19]. For instance, Barbot and Kaufman [9] discovered that participants' empathy levels were significantly amplified by the immersion and

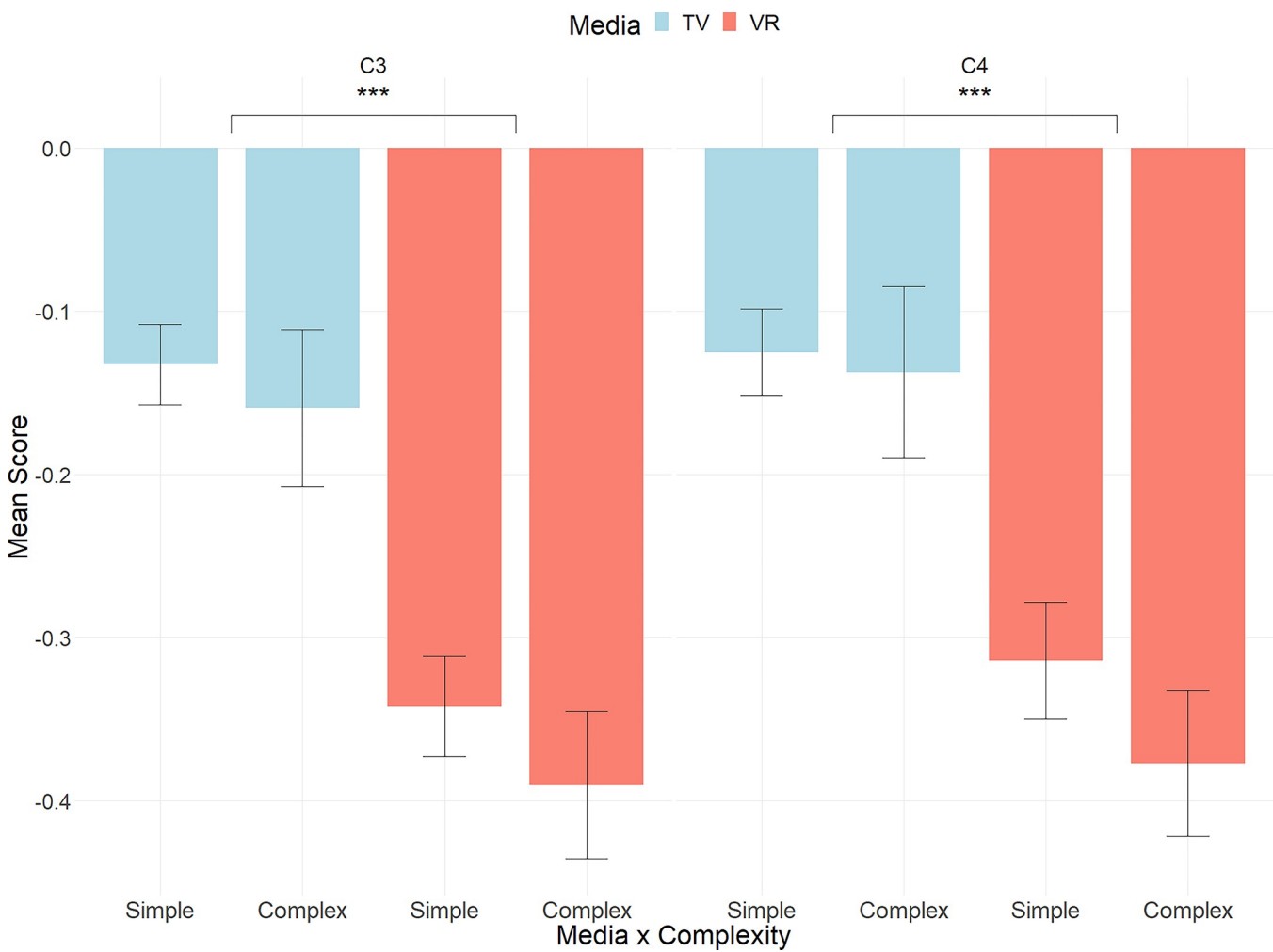

**Fig 1. Mu rhythm suppression (8 to 13 Hz) for each electrode (C3, C4) across all conditions (media x complexity).**

presence experienced in diverse immersive VR scenarios, gauged through self-report surveys. Nelson et al. [15] noted an increase in pro-environmental actions such as donating to charity among participants after they engaged with a VR presentation highlighting underwater environmental threats. Nevertheless, despite the significance of these findings, it is worth noting they largely relied on self-report measures and behavioural indices. Although these methodologies offer valuable insights, they are fundamentally subjective and might not fully capture the intricacy of empathic responses. In contrast, our study, through the use of EEG, offers a more objective and direct insight into the neural mechanisms underpinning these empathic experiences in different media environments. EEG's high temporal resolution enables us to observe immediate neural responses, which may be overlooked or misinterpreted by subjective behavioural measures. This approach offers a more nuanced understanding of empathic engagement dynamics and contributes to the existing body of research by connecting behavioural findings with their neurophysiological correlates in VR-induced empathy.

Furthermore, the results of this study were not confined to specific action types such as grasping, given the absence of an interaction between media type and action complexity. The short video clips utilised in this research depicted simple actions of object grasping, following

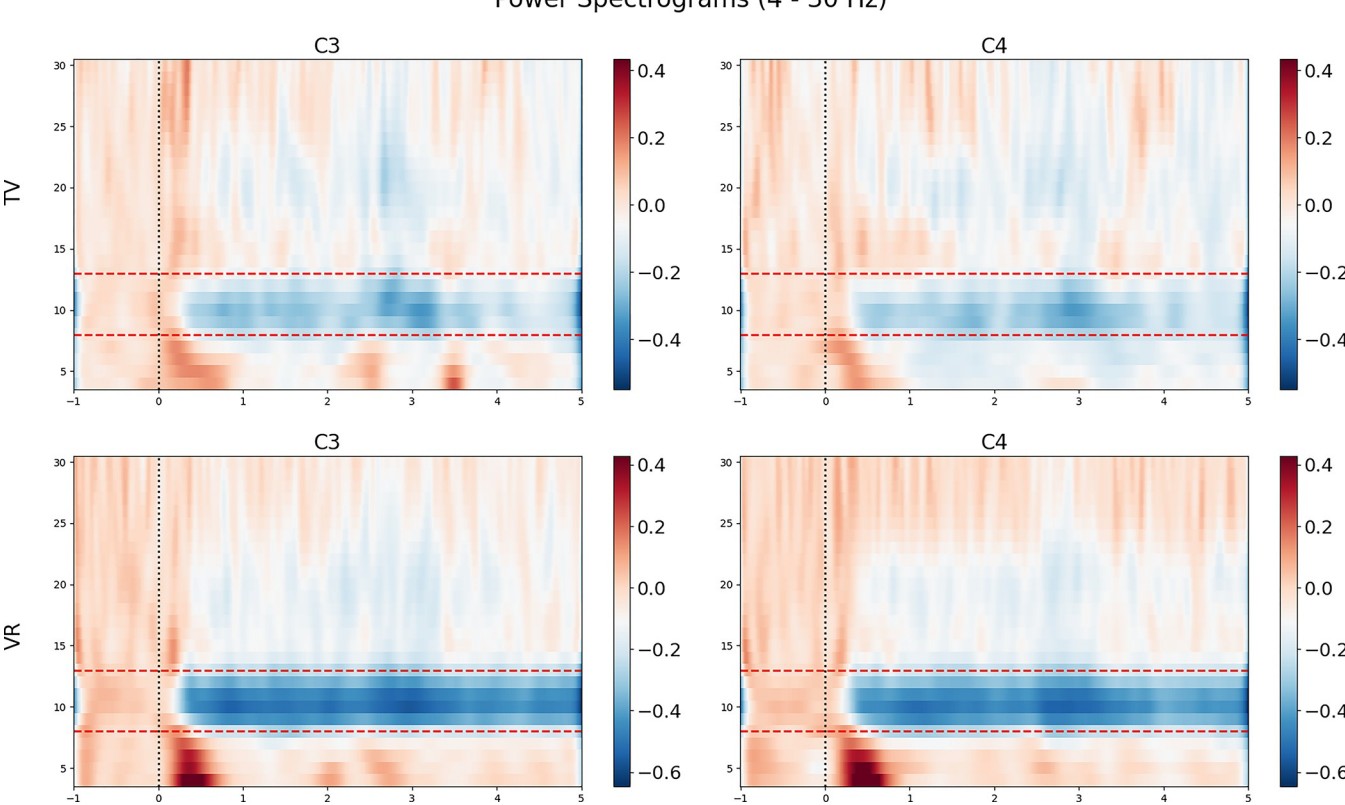

**Fig 2. Power spectrograms ranging from 4 to 30 Hz.** Areas of power decrease and increase are indicated by blue and red shading, respectively. The mu rhythm range is marked by a red dotted line.

conventional material used in mu rhythm suppression studies. However, the inherent simplicity of these actions might impose limitations on the interpretability of the outcomes because mu rhythm suppression may only be associated with certain types of actions. Particularly, such simple actions might not represent those that typically evoke empathetic responses in the media narratives under investigation. To address this limitation, in the current study, we

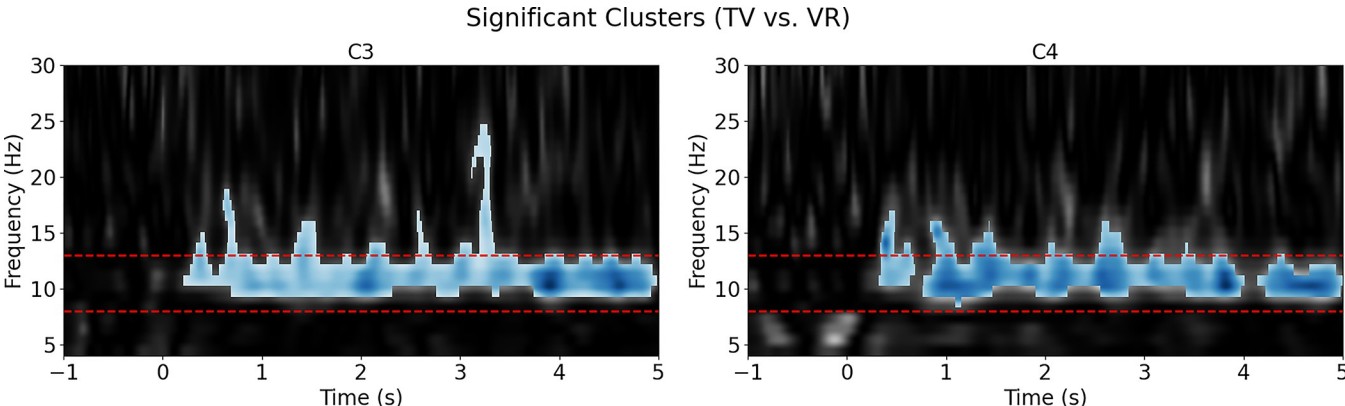

**Fig 3. Nonparametric permutation F-test results for media condition (VR vs. TV).** Shaded areas signify significant clusters, with blue indicating greater suppression in VR conditions and red indicating greater enhancement in VR conditions.

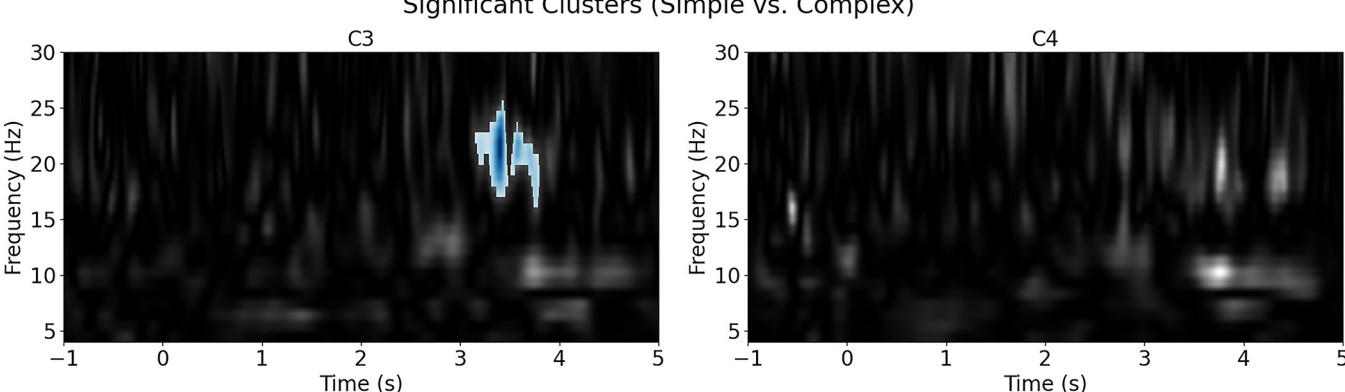

**Fig 4. Nonparametric permutation F-test results for complexity condition (simple vs. complex).** Shaded areas signify significant clusters, with blue indicating greater suppression in complex conditions and red indicating greater enhancement in complex conditions.

incorporated actions of varied complexity, and the findings indicated the type of action exerted no influence on the outcomes because more complex actions also prompted mu rhythm suppression. We do recognise, however, that the complex actions incorporated in this study are not fully representative of the intricate action sequences often found in media narratives. The results of our study may not be directly applicable to scenarios of greater complexity, thus warranting further research into this matter. Nonetheless, we endeavoured to gauge the potential generalisability of this simple material and at least demonstrated that these responses are not limited to certain types of actions.

One might argue that there was no difference in mu rhythm suppression between simple and complex conditions merely because the level of complexity was not sufficiently different. The increased number of consecutive actions and the involvement of the entire body might not suffice to markedly increase the complexity of an action. Such an argument is, however, partially countered by the differences observed in the beta band. The results revealed an increased beta suppression in the complex conditions. Beta suppression, often observed concurrently with mu rhythm suppression, is known to be associated with motor execution or observation [51, 52, 65–70]. One characteristic that sets beta rhythms apart from mu rhythms within the alpha frequency band is the beta rebound, or a transient increase in the power of the beta frequency band after a motor action is completed [67, 68, 71–76], which could potentially account for the observed discrepancy in beta suppression. The brevity of the simple actions featured in our study may have resulted in an earlier completion of the actions, leading to an earlier rebound effect. Additionally, beta power is known to be influenced by certain motor parameters, such as the velocity and perspective of the action [50–52, 70]. Given that the simple and complex actions differed in velocity and perspective, they might have contributed to the observed differences in beta suppression. However, considering that primary objective of this study was not to directly compare simple and complex actions, various factors potentially affecting beta suppression were not meticulously regulated. Consequently, it is not possible to fully account for the difference between simple and complex actions. Nonetheless, our findings do indicate the presence of differences between these two types of actions, suggesting that the findings of this study might permit a broader interpretive scope, particularly in relation to actions of greater complexity.

What specific aspects of VR media might have elicited greater empathetic responses compared to TV, as evidenced by our findings? This study operated under the hypothesis that the immersive nature of VR is responsible for these empathetic responses. The immersive quality

of VR is a result of the interplay of multiple factors. These include the perceived control over the virtual environment afforded by VR's interactivity, the realism derived from multi-sensory inputs such as haptic feedback, three-dimensional visuals, and spatial audio, as well as the perception of being in a new space that emerges when users are separated from the external environment [23, 24, 26, 77, 78]. Collectively, these factors contribute to the creation of a compelling and immersive virtual environment. Among these various factors, the aspect deemed to have notably influenced the outcomes of this research is the sense of immersion stemming from isolation from external surroundings. When a user is isolated from their physical environment, their attention becomes more focused on the virtual world. This isolation, typically achieved by using VR headsets that block out external sights and sounds, helps create a more convincing and enveloping experience. By minimizing distractions from the real world, VR can more effectively trick users into perceiving it is in a different environment, enhancing the overall sense of presence and immersion within the virtual space [23, 77–80]. In this study, the videos presented through VR and TV were identical in terms of video quality, audio, three-dimensionality, controllability, and interactivity. Therefore, the most prominent difference in the immersive qualities of VR between the two conditions was that wearing VR equipment isolated the participants from their surrounding environment. While participants in the TV condition had to remain aware of their physical surroundings, those in the VR setup were isolated from the external experimental environment, enabling concentrated engagement with the video. This divergence may have heightened the immersive experience in the VR setting, potentially leading to increased empathetic responses.

Nevertheless, it is important to approach this interpretation with consideration for two aspects. Firstly, the role of isolation in enhancing empathetic responses is inferred from the conditions of our experiment rather than a direct investigation into the effects of separation from external environments. Our research primarily focused on comparing TV and VR, ensuring that all video elements, except the media devices, were uniformly controlled. Hence, observed differences between the conditions are likely attributable to inherent disparities in the media themselves. The blocking of external environments, a pronounced difference induced by the media choice, might be interpreted as a crucial factor. However, as the present study did not explicitly experiment with isolation per se, comprehensive exploration of isolation's impact necessitates further research. For instance, a study comparing the isolating effect could involve a TV setting engineered to block external distractions, akin to VR equipment, allowing concentrated focus on the monitor. The second consideration is that mere sensory isolation from external environments does not guarantee detachment and immersion. The use of VR gear, while focusing the user's attention within the virtual environment, does not preclude awareness of the external world. For instance, wearing VR in public spaces [81] or being aware of others' presence in the same room [82], may provoke anxiety and disrupt the immersive experience. In this study, conducted in the relatively secure setting of an experimental booth, the participants were less likely to experience insecurity or a heightened awareness of the external world. Nevertheless, it is crucial to recognize that the immersion resulting from this isolation is not solely due to sensory blocking. If the existence of the outside world is clearly perceived despite being sensually isolated, it could rather impede the immersive process.

Our study provides neurophysiological evidence indicating that VR elicits stronger empathic responses compared to traditional TV in the context of our experiment, contributing to the ongoing discourse on the potential of VR as an empathy-enhancing medium. The results consistently demonstrated a more significant mu rhythm suppression in VR conditions as opposed to TV conditions. Notably, the suppression of the mu rhythm was not confined to simple action but also extended to complex actions, which suggests our observations may be

applicable to actions of greater complexity. We propose that the immersive quality of VR, particularly its ability to isolate users from their external environment, can be interpreted as a critical factor distinguishing it from TV in enhancing empathy. However, these findings still necessitate further research. Future studies should aim to explore a wider range of actions, including more complex and nuanced actions, to better reflect the varied action sequences found in real-world media. Moreover, additional research is required to fully understand how isolation or other immersive characteristics contributes to the empathic experience in VR. Such research would contribute to a more comprehensive understanding of how different types of media, the actions depicted therein, and immersive qualities influence empathic responses. Additionally, exploring these dynamics in other immersive media, beyond VR, could provide further insight into the role of media immersion in empathic engagement. This could also help inform the development of more effective and engaging media content, whether for entertainment, educational, therapeutic, or other purposes. Last, future research should aim to explore these findings in more diverse populations, including varying age groups, cultural backgrounds, and socioeconomic statuses, to enhance the generalizability and understanding of our study's implications across a wider array of demographic groups.

In concluding our discussion, we carefully highlight the contributions of our study from two crucial perspectives: First, our findings advance the field by elucidating the complex dynamics of empathy in VR environments. By demonstrating the neurophysiological underpinnings of empathy through Mu rhythm analysis, our study fills a crucial gap in understanding and substantiates VR's efficacy as a potent medium for empathy enhancement. This contribution is pivotal, offering a scientifically grounded perspective on leveraging VR to foster deeper empathic engagement. Second, exploring the neurophysiological responses associated with media immersion opens new horizons for academic exploration, laying a solid foundation for future research to build upon. By identifying specific neural correlates of empathic engagement in VR, our work invites a multidisciplinary dialogue and encourages further investigations that could explore various dimensions of empathy in immersive media. Thus, our study not only enriches the existing discourse with insights and empirical evidence but also sets the stage for a continued and expanding exploration into the capacity of immersive technologies to shape human empathy, ensuring the conversation remains dynamic and progressively forward-looking.

## Author Contributions

**Conceptualization:** Jong-Hyun Lee, Young-Sung Kwon.

**Data curation:** Jong-Hyun Lee.

**Formal analysis:** Jong-Hyun Lee.

**Funding acquisition:** Young-Sung Kwon.

**Investigation:** Sung Eun Lee.

**Methodology:** Jong-Hyun Lee, Sung Eun Lee.

**Project administration:** Sung Eun Lee, Young-Sung Kwon.

**Software:** Jong-Hyun Lee.

**Supervision:** Young-Sung Kwon.

**Validation:** Young-Sung Kwon.

**Writing – original draft:** Jong-Hyun Lee.

**Writing – review & editing:** Young-Sung Kwon.

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
