## [Decision Letter · Decision Letter 0]

24 Oct 2023

PONE-D-23-28605Exploring Empathic Engagement in Immersive Media: An EEG study on Mu rhythm suppression in VRPLOS ONE

Dear Dr. Kwon,

Thank you for submitting your manuscript to PLOS ONE. After careful consideration, we feel that it has merit but does not fully meet PLOS ONE’s publication criteria as it currently stands. Therefore, we invite you to submit a revised version of the manuscript that addresses the points raised during the review process. Major revision is required as per the feedback and comments from the reviewers (feedback appended below).

We look forward to receiving your revised manuscript.

Kind regards,

Umer Asgher, PhD

Academic Editor

PLOS ONE

Journal Requirements:

Reviewers' comments:

Reviewer's Responses to Questions

**Comments to the Author**

1. Is the manuscript technically sound, and do the data support the conclusions?

Reviewer #1: Partly

Reviewer #2: Yes

Reviewer #3: Yes

2. Has the statistical analysis been performed appropriately and rigorously? 

Reviewer #1: I Don't Know

Reviewer #2: Yes

Reviewer #3: Yes

3. Have the authors made all data underlying the findings in their manuscript fully available?

Reviewer #1: No

Reviewer #2: Yes

Reviewer #3: Yes

4. Is the manuscript presented in an intelligible fashion and written in standard English?

Reviewer #1: Yes

Reviewer #2: Yes

Reviewer #3: Yes

5. Review Comments to the Author

Reviewer #1: Dear Authors,

Thank you for providing me with this opportunity to review the manuscript titled ”Exploring Empathic Engagement in Immersive Media: An EEG study on Mu rhythm suppression in VR”. Hopefully this review statement helps in developing the manuscript further.

To begin with, the introduction could position the manuscript more explicitly vis-à-vis prior literature. For instance, why do we need to compare VR and TV? Furthermore, the argument “empathy…is a crucial element in media experiences” needs more flesh around the bones. How is it a crucial element? What do we know about it so far? In addition, “…for eliciting empathic responses…” (p. 1) would benefit from concrete examples.

In a way, it feels as if the manuscript jumps directly to the topic without properly paving the way. In other words, rushing to the topic results in the manuscript being inadequately positioned with regards to prior literature. As such, the introduction should provide a more nuanced and granulated image of the current body of knowledge.

Methodology: based on what criteria / prior research were the videos created? Also, were the actors shown in the videos? If so, were they shown in full or partially? Stills of the videos would be very helpful. Also, could you cite similar studies to strengthen the methodologocial choices? What does the simple / complex distinction tell us?

Furthermore, engagement towards what? Actor, object, action?

Were the VR and TV clips similar / identical?

Is it possible to share the material so others could replicate your study? Did the author team create the videos or someone else?

Methods section pp. 8-10 does not seem to have any references.

Point being, the methodology section needs to be stronger and more transparent. As it is now, challenging to see how insights were derived from the data.

Findings:

Is it possible to have a more nuanced treatment of the findings? For example, were there differences across the videos (not only simple – complex distinction)? As it is now, the findings feel very flat and after reading the manuscript I was hoping to learn more rather than “VR is superior to TV”; yes, I buy this, but how exactly could VR be more superior in this context? Under what conditions? Is VR always more superior?

Conversely: my co-author and I studied the use of VR in a public space (metro) and one of our findings was that people felt a bit insecure with a VR headset on. In other words, they weren’t always sure they were safe. While we didn’t compare VR with TV, one could say in public spaces VR might not be as efficient as TV because of the safety / security aspect. So, a more nuanced analysis of the findings would definitely make the paper stronger.

For instance, p. 15: “…a more nuanced understanding of empathic engagement dynamics…” – yes, this would be great, yet the manuscript does not address such dynamics at the moment. This would be a great vantage point for making contributions with this study to the current body of knowledge.

Is p. 16 necessary? How does it advance the contributions of the manuscript? Granted, I am definitely not an expert on beta suppression, but regardless this section feels a bit like a sidestep. Instead, I would focus more on the above rather than introducing new concepts.

p. 17 – “…VR may evoke stronger empathic responses” – this is super interesting and yet another potentially beneficial vantage point. How could others build on these insights?

Building on the above, and repeated below, it would be great to highlight this vantage point: “how could this paper open up rather than close down discussion?”. How might one build on this work? Instead of reporting findings, present them so that the manuscript explicitly encourages others to do more work in this domain. Here, a helpful vantage point could be this: what might be the three things the reader could take away from this manuscript?

Once again, thank you for the opportunity to review this manuscript. Hope this review statement helps in developing the manuscript further.

Minor comments:

- After revising the manuscript, the abstract could also be revised so it does not oversell the findings. There is no need for that. Instead, focus on the contributions and how these findings open up rather than close down discussion

- P. 3 – “Empathy, encompassing both…” sentence repeats what has been said above

- P. 3 – last paragraph is very interesting and promising

- P. 4 – “Immersion captivates…” – could be rephrased; how exactly does immersion lead to emotional connection?

- P. 4 – “For instance, in cinema…” – not sure the causality presented in this sentence is clear. Either tone down or explicitly explain the situation to the reader

- P. 4 – top paragraph – in general, the top part presents strong causalities that, in fact, are not necessary from the paper’s contributions’ point of view. Instead, provide a more nuanced treatment of the current body of knowledge so the paper’s contributions stand out on their own.

- P. 5 “ultimate empathy machine” – similarly, a very strong statement. No need to oversell the topic.

- P. 5 – Clouds Over Sidra – this is a good example, yes, but are there more examples?

- P. 5 middle paragraph is very nice as it has a nuanced and granular take on prior literature

- P. 6 – top paragraph – examples would make this paragraph more accessible

- P. 17 – “…superiority of VR…” – yet another strong statement

Reviewer #2: The manuscript is well-organized and clearly written, with a well-defined research question and hypothesis. The methods are rigorous and appropriate for addressing the research question, and the results are clearly presented and supported by the data. I particularly appreciate the thoroughness of the discussion section, which provides a comprehensive analysis of the findings and their implications.

Overall, I believe that the manuscript is a valuable contribution to the field, and I am confident that it will be well-received by the readership of PLOS ONE. I look forward to seeing it published.

Reviewer #3: A review of the manuscript titled "Exploring Empathic Engagement in Immersive Media: An EEG study on Mu rhythm suppression in VR."

The introduction effectively contextualizes the research, outlines the research questions, highlights the significance of the study, and provides a clear hypothesis. It also demonstrates a solid connection to existing literature and a comprehensive understanding of the topic, making it a well-structured and informative paper section.

The front matter provides a clear and informative background on immersive media technologies, especially virtual reality (VR), and their potential impact on empathy. It effectively sets the stage for the research study by outlining the significance of empathy in media experiences. The paper includes citations from prominent media studies and empathy scholars; therefore, the research is grounded in existing literature and theoretical frameworks.

The paper effectively highlights a research gap related to the effectiveness of VR in inducing empathy and the lack of neurophysiological evidence in this domain. It then articulates the central research questions. The discussion on the relationship between immersion and empathy is well-founded and coherent. It explains how immersion in various media forms can foster empathic engagement and sets the stage for the unique immersive qualities of VR.

The authors discuss the distinctive characteristics of VR, such as its capacity to create a sense of presence and interactivity, effectively underscores why VR is of particular interest in the study of empathy and cites previous research to support claims about VR's immersive qualities — adding credibility to the later discussion. The paper acknowledges the existence of mixed results in earlier research regarding VR's impact on empathy — demonstrating an awareness of the complexity of the topic and a willingness to critically assess the claims made about VR as the "ultimate empathy machine."

The introduction proposes the neurobiological basis of empathy and its connection to the mirror neuron system — adding depth to the discussion and positioning the research within the field of neuroscience. The explanation of the mu rhythm and its connection to empathic engagement is well-defined. It explains how EEG data can be used as an empirical measure of empathy, a vital aspect of the study's methodology.

The introduction concludes with a clear hypothesis that states the expected outcome of the study, which is that VR will lead to a more robust empathic response compared to TV and that the complexity of actions may modulate this response.

The Methods section outlines the comprehensive and well-structured research design, data collection, and analysis procedures. The section begins with a clear and concise description of the participants, including their demographics and the recruitment method. Including gender and age information and the number of participants is crucial for transparency. Mentioning that the experiment was conducted per the university's ethics guidelines reinforces the ethical considerations.

The description of the experimental stimuli is detailed, including information about the camera used and the source of the actors. Categorizing stimuli into simple and complex actions adds clarity to the design. However, providing information on the number of clips in each category would be beneficial to understanding the balance in stimuli representation.

The procedure section offers a step-by-step explanation of how the experiment was conducted. It covers participant instructions, EEG data collection, and the overall time frame of the investigation. The random assignment of participants to TV and VR conditions and the provision of breaks between states is essential for minimizing order effects and ensuring participant comfort. The compensation details are also evident.

The description of the equipment used for the TV and VR conditions is detailed and adequately explains how the VR headset and EEG devices were combined without interference — this is vital information for understanding how the experiment was conducted. The section clearly describes the EEG recording setup, including the type of EEG recorder, electrode placement, and electrode reference. The explanation of how the VR headset was combined with EEG devices is well-detailed and suggests care in the experimental setup.

The preprocessing steps are thoroughly explained, from filtering to artifact rejection. Including the Python packages and the description of the data loss due to artifact rejection adds transparency to the methodology. However, mentioning the specific criteria for data rejection might be helpful to address potential concerns about subjective judgments.

The description of the time-frequency analysis is concise and mentions the choice of electrodes, frequency range, and power normalization. These details provide insight into the analytical approach. The section explains the statistical methods employed, including repeated measures ANOVA and non-parametric permutation F-test. It mentions the handling of multiple comparisons and correction methods, enhancing the rigor of the analysis.

The Results section presents the findings, including the statistical tests performed and the graphical representations of the data. The area begins with the results of the repeated measures ANOVA, which is appropriate for testing main effects and interactions. The main impact of media on both C3 and C4 channels is significant, with p-values provided — this demonstrates that the media condition (VR vs. TV) has a notable impact on mu rhythm suppression in the specified EEG channels. The significance level is appropriately indicated (p < 0.001). The results effectively demonstrate that the results from the non-parametric permutation F-test are consistent with the ANOVA findings — this adds credibility to the results by showing consistency across two statistical methods. The report also notes the absence of significant clusters in the complexity analysis, which is essential information for the reader. It highlights that no significant differences in mu rhythm suppression were found between simple and complex conditions. The significance threshold (p<0.025) is clearly stated. The inclusion of figures (Figure 1, Figure 2, and Figure 3) helps visualize the results. Figure 1 provides a graphical representation of the mu rhythm suppression across conditions, assisting the readers to interpret the main effect of media and complexity. Figure 2 offers power spectrograms, which help understand changes in power across the frequency range. Figure 3 includes a clear and informative legend that explains the color coding used to indicate significant clusters. The distinction between blue and red shading for more significant suppression or enhancement is presented, making the results easier to interpret. Still, the color coding needs to be changed for color-blind accessibility.

The paper's Discussion section effectively addresses the findings and provides insightful interpretations and implications. The section provides a clear and comprehensive understanding of the study's results. It appropriately discusses the main conclusions of mu rhythm suppression in VR versus TV conditions. The discussion effectively links the results to previous immersive media, empathy, and mirror neuron system activation research. This connection to existing literature adds depth and context to the findings. The paper correctly highlights the advantages of using EEG to measure neural responses. The discussion points out the benefits of objective, neurophysiological measures in contrast to subjective self-report measures often used in previous studies. The section addresses the role of action complexity in the findings and raises valuable points about the limitations of using simple actions in the study. The discussion acknowledges the potential limitations and opens the door for future research.

The discussion of beta suppression and its potential implications for the observed differences between simple and complex conditions is well-presented. It appropriately suggests that future research should delve deeper into the influence of action complexity on neural responses. This section successfully summarizes the study's main contributions to the field, emphasizing the potential of VR as an empathy-enhancing medium. It highlights the consistent mu rhythm suppression observed in VR and the need for further research to explore different media types and actions.

The paper concludes with insightful suggestions for future research, including exploring action complexity, the role of media immersion in empathic engagement, and potential applications in various domains. The final section is well-structured and effectively interprets the results in the context of existing literature. It also identifies limitations and provides valuable directions for future research, making it a solid and informative part of the paper.

The paper demonstrates a clear and well-structured format, making it easy for readers to follow the research design, methods, findings, and interpretations. The article effectively integrates relevant literature into the introduction and discussion sections, enhancing the study's context and relevance. Using EEG as a neurophysiological measure is appropriate, and the paper provides a good rationale for this choice. It highlights the advantages of using objective criteria compared to subjective self-report measures. The report explores an exciting and relevant topic by investigating immersive media's impact on empathic experiences, particularly VR. This research area has practical applications and implications for various fields. The methods section is well-detailed, and using both ANOVA and non-parametric permutation F-tests provides robust statistical analysis. The paper addresses participant demographics, materials, apparatus, and data analysis factors. The "Discussion" section is a strength of the article. It effectively interprets the results, connects them to existing literature, acknowledges limitations, and suggests promising directions for future research.

While the paper demonstrates several strengths, some areas could be improved or have limitations.

• The sample size in the study is relatively small (30 participants), which may limit the generalizability of the findings. Increasing the sample size could strengthen the study's reliability and validity.

• The study mentions recruiting university students, which might introduce a potential bias in the sample's demographic characteristics. More information on participant demographics, such as age, background, and experience with VR, could be helpful.

• The paper does not mention diversity in the sample, which is a common concern in research. It's essential to consider the impact of gender, age, and cultural background on empathic responses.

• The paper acknowledges the limitation of using simple actions for the study. Exploring a broader range of activities, including those more typical of complex media narratives, would be valuable — for example, the 2021 paper Exploring Virtual Reality for Quality Immersive Empathy Building Experiences in Behaviour & Information Technology.

• The study's findings are based on specific experimental conditions, and the paper acknowledges that more research is needed. It's important to highlight that the results may not universally apply to all VR or media content.

• The study is conducted in a controlled laboratory setting, and the actions in the videos are artificial. It may not fully represent the complexity and emotional engagement of real-world media experiences.

• While the study looks at neural markers associated with empathy, it's essential to recognize that empathy is a complex psychological and neurobiological phenomenon. The paper acknowledges this but should emphasize the complexity of the topic further.

• The paper mentions obtaining informed consent and following ethical guidelines. Still, more information on the ethical aspects of the research, particularly in the context of using VR technology, would be beneficial.

• The paper's conclusion could be more concise and emphasize the essential findings and implications.

It's important to note that many of these concerns are typical of scientific research and should be considered opportunities for further investigation and improvement rather than significant flaws.

In summary, the paper demonstrates a robust research design, a clear presentation of results, and insightful interpretations. It effectively contributes to understanding the relationship between immersive media, empathy, and neural mechanisms. Overall, the paper appears to be of good quality and contributes meaningfully to its field of study.

6. PLOS authors have the option to publish the peer review history of their article (what does this mean?). If published, this will include your full peer review and any attached files.

Reviewer #1: **Yes: **Miikka J. Lehtonen

Reviewer #2: **Yes: **Wahab Khan

Reviewer #3: **Yes: **Gareth W. Young

---

## [Author Response · Author response to Decision Letter 0]

23 Feb 2024

Dear reviewers,

We would like to thank you for the detailed and insightful comments. In response, we have made revisions that we feel much strengthened our manuscript. Please see detailed responses to your individual comments in the attached 'Response to Reviewers' file. We have so far tried our best to revise our manuscript and clarify important concerns that you raised. Please do not hesitate to let us know if you need more information or have other questions or concerns. Much gratitude again for your time and contribution to this work. 

Respectfully yours, 

Author

---

## [Decision Letter · Decision Letter 1]

26 Mar 2024

PONE-D-23-28605R1Exploring Empathic Engagement in Immersive Media: An EEG study on Mu rhythm suppression in VRPLOS ONE

Dear Dr. Kwon,

Thank you for submitting your manuscript to PLOS ONE. After careful consideration, we feel that it has merit but does not fully meet PLOS ONE’s publication criteria as it currently stands. Therefore, we invite you to submit a revised version of the manuscript that addresses the points raised during the review process. Please submit your revised manuscript by May 10 2024 11:59PM. If you will need more time than this to complete your revisions, please reply to this message or contact the journal office at plosone@plos.org. Please include the following items when submitting your revised manuscript:A rebuttal letter that responds to each point raised by the academic editor and reviewer(s). You should upload this letter as a separate file labeled 'Response to Reviewers'.A marked-up copy of your manuscript that highlights changes made to the original version. You should upload this as a separate file labeled 'Revised Manuscript with Track Changes'.An unmarked version of your revised paper without tracked changes. You should upload this as a separate file labeled 'Manuscript'.If applicable, we recommend that you deposit your laboratory protocols in protocols.io to enhance the reproducibility of your results. Protocols.io assigns your protocol its own identifier (DOI) so that it can be cited independently in the future. For instructions see: https://journals.plos.org/plosone/s/submission-guidelines#loc-laboratory-protocols. Additionally, PLOS ONE offers an option for publishing peer-reviewed Lab Protocol articles, which describe protocols hosted on protocols.io. Read more information on sharing protocols at https://plos.org/protocols?utm_medium=editorial-email&utm_source=authorletters&utm_campaign=protocols.

We look forward to receiving your revised manuscript.

Kind regards,

Umer Asgher, PhD

Academic Editor

PLOS ONE

Journal Requirements:

Reviewers' comments:

Reviewer's Responses to Questions

**Comments to the Author**

1. If the authors have adequately addressed your comments raised in a previous round of review and you feel that this manuscript is now acceptable for publication, you may indicate that here to bypass the “Comments to the Author” section, enter your conflict of interest statement in the “Confidential to Editor” section, and submit your "Accept" recommendation.

Reviewer #1: (No Response)

Reviewer #3: (No Response)

2. Is the manuscript technically sound, and do the data support the conclusions?

Reviewer #1: Yes

Reviewer #3: Yes

3. Has the statistical analysis been performed appropriately and rigorously? 

Reviewer #1: Yes

Reviewer #3: Yes

4. Have the authors made all data underlying the findings in their manuscript fully available?

Reviewer #1: Yes

Reviewer #3: No

5. Is the manuscript presented in an intelligible fashion and written in standard English?

Reviewer #1: Yes

Reviewer #3: Yes

6. Review Comments to the Author

**Reviewer #1:** Dear Authors,

I was delighted to engage with this revised manuscript titled “Exploring Empathic Engagement in Immersive Media: An EEG study on Mu rhythm suppression in VR”. Most of the issues addressed by the reviewers have now been resolved, and many thanks for providing such detailed responses to the review statements.

At this point, there are only two minor issues and one tiny comment:

Introduction (minor issue): while this section has considerably improved, there is room for making it even stronger to help the readers engage with the manuscript more thoroughly. For instance, this is how the introduction’s structure could look like:

1. Engagement has been found to be an issue in immersive media

2. Empathy, amongst other factors such as immersion, plays a crucial role in evoking engagement

3. Yet, we do not know much about how, and through what mechanisms, VR evokes empathic engagement

4. Thus, more research is required to better understand how empathic engagement in VR differs from more traditional media

The above is just a suggestion, mind you. The rest of the manuscript is now very strong, and getting the introduction to the same level would be very critical at the moment. To be precise, I do not see this as a major undertaking; all the key ideas are there in the introduction, it just needs a stronger structure.

Methodology (tiny comment): Were the participants given USD 24 for their participation or X won equivalent to USD 24? (Also, “$24 USD” should be either $24 or USD 24) As said, tiny comment, but crucial in the sense of ensuring the manuscript does not contain any ambiguities. As a concrete suggestion, I would disclose how many won each participant received and in brackets how much it would be in USD. This, of course, assuming that the participants received won instead of USD. If they did receive USD, my apologies for the wrong assumption. In such case, please only consider the comment on the spelling format.

Contributions (minor issue): at the end of the discussion, I would highlight the contributions more explicitly from two perspectives: 1) how do the findings contribute to extant body of knowledge, and 2) how do the findings open up rather than close down conversation (i.e. future research avenues). Again, all the information is there, I would just make these two aspects very explicit to the reader so that the manuscript would receive the attention and engagement it deserves.

Once again, thank you for this review opportunity. Revisions have considerably strengthened the manuscript, nicely done.

**Reviewer #3:** The newest version of the paper "Exploring Empathic Engagement in Immersive Media: An EEG study on Mu rhythm suppression in VR" significantly clarifies how it contributes to understanding the neural mechanisms of empathic engagement in immersive media. Its methodological rigor, innovative approach, and insightful analysis are commendable. However, addressing the areas highlighted below for improvement could enhance the findings' robustness and impact, paving the way for further research in this intriguing field.

The paper employs an innovative EEG approach to measure mu rhythm suppression to indicate empathic engagement in immersive media — this provides a quantifiable measure of empathy, offering valuable insights into the neural underpinnings of empathic responses in VR and TV media environments. The results indicate a more significant mu rhythm suppression in VR than in TV, suggesting a more robust empathic response in immersive VR environments — this contributes to the growing literature on VR's potential as an empathy-enhancing medium.

The detailed and well-structured methodology section provides clear information on participant selection, materials, procedures, apparatus, EEG recording, and preprocessing. This thoroughness ensures reproducibility and enhances the study's credibility. While the study attempts to address the complexity of actions depicted in VR and TV media, it acknowledges that the complex actions used may not fully represent the range of actions in media narratives. Future studies could explore a more comprehensive array of complex actions to enhance understanding of empathic responses in varied contexts. The paper could further elaborate on the technical aspects of EEG data collection and analysis, including the choice of electrodes, the rationale behind the specific frequency ranges examined, and any potential limitations these choices entail.

The statistical analyses are robust, using repeated measures ANOVA and non-parametric permutation F-tests to examine the data. This dual approach strengthens the study's findings by addressing the potential methodological limitations of each statistical method. The discussion suggests that the isolation effect of VR (i.e., blocking out the external environment) may contribute to enhanced empathic responses. However, this is inferred rather than directly investigated. Future research examining the role of isolation in empathy could provide more definitive insights.

The discussion provides a thoughtful interpretation of the findings, linking them to existing research and theory. It also acknowledges the study's limitations and suggests avenues for future research, demonstrating the authors' critical engagement with their work. The paper could benefit from a broader discussion on the generalizability of its findings. For example, the sample size and demographics (e.g., all participants being university students) may limit the applicability of the results to broader populations.

While the paper posits that the immersive quality of VR is primarily responsible for enhanced empathic responses, it would be beneficial to consider and discuss alternative explanations or contributing factors more thoroughly. For instance, the role of narrative engagement, user interactivity, and the novelty of VR technology could also influence empathic responses.

7. PLOS authors have the option to publish the peer review history of their article (what does this mean?). If published, this will include your full peer review and any attached files.

Reviewer #1: **Yes: **Miikka J. Lehtonen

Reviewer #3: **Yes: **Gareth W. Young

---

## [Author Response · Author response to Decision Letter 1]

31 Mar 2024

Dear reviewers,

We want to thank you once again for the detailed and insightful comments. We have tried our best to revise our manuscript and clarify important concerns that you raised. Please see detailed responses to your individual comments on the 'Response to Reviewers (Minor Revision)' file. Again, we are grateful for your time and contribution to this work.

Sincerely,

Corresponding author

---

## [Decision Letter · Decision Letter 2]

29 Apr 2024

Exploring Empathic Engagement in Immersive Media: An EEG study on Mu rhythm suppression in VR

PONE-D-23-28605R2

Dear Dr. Kwon,

We’re pleased to inform you that your manuscript has been judged scientifically suitable for publication and will be formally accepted for publication once it meets all outstanding technical requirements.

Kind regards,

Umer Asgher, PhD

Academic Editor

PLOS ONE

Additional Editor Comments (optional):

Reviewers' comments:

Reviewer's Responses to Questions

**Comments to the Author**

1. If the authors have adequately addressed your comments raised in a previous round of review and you feel that this manuscript is now acceptable for publication, you may indicate that here to bypass the “Comments to the Author” section, enter your conflict of interest statement in the “Confidential to Editor” section, and submit your "Accept" recommendation.

Reviewer #1: All comments have been addressed

2. Is the manuscript technically sound, and do the data support the conclusions?

Reviewer #1: Yes

3. Has the statistical analysis been performed appropriately and rigorously? 

Reviewer #1: Yes

4. Have the authors made all data underlying the findings in their manuscript fully available?

Reviewer #1: Yes

5. Is the manuscript presented in an intelligible fashion and written in standard English?

Reviewer #1: Yes

6. Review Comments to the Author

Reviewer #1: Dear Authors,

Thank you for diligently addressing the comments from both reviewers. The manuscript reads now very smoothly, and it is also nicely grounded in prior literature. More importantly, however, I am convinced that this paper will inspire readers to carry out further inquiries. Well done!

Thank you for such a pleasurable review process. Already now I have learned a lot from this manuscript. Also, thank you for very constructive and kind responses to reviewers’ comments. It has been a pleasure working with you on this paper.

7. PLOS authors have the option to publish the peer review history of their article (what does this mean?). If published, this will include your full peer review and any attached files.

Reviewer #1: **Yes: **Miikka J. Lehtonen

---

## [Editor Report · Acceptance letter]

7 May 2024

PONE-D-23-28605R2 

PLOS ONE

Dear Dr. Kwon, 

I'm pleased to inform you that your manuscript has been deemed suitable for publication in PLOS ONE. Congratulations! Your manuscript is now being handed over to our production team.

Kind regards, 

on behalf of

Dr. Umer Asgher 

Academic Editor

PLOS ONE